# Immune-Related Pneumonitis Was Decreased by Addition of Chemotherapy with PD-1/L1 Inhibitors: Systematic Review and Network Meta-Analysis of Randomized Controlled Trials (RCTs)

Yi-Xiu Long [1], Yue Sun [2], Rui-Zhi Liu [3], Ming-Yi Zhang [1], Jing Zhao [1], Yu-Qing Wang [1], Yu-Wen Zhou [1], Ke Cheng [1], Ye Chen [1], Cai-Rong Zhu [2] and Ji-Yan Liu [1,4,5,*]

[1] Department of Biotherapy, Cancer Center, West China Hospital of Sichuan University, Chengdu 610041, China; longyixiu2021@163.com (Y.-X.L.); zhangmingyi319@163.com (M.-Y.Z.); greencandy1216@163.com (J.Z.); 2020224020143@stu.scu.edu.cn (Y.-Q.W.); drzhouyuwen@163.com (Y.-W.Z.); chengke@wchscu.cn (K.C.); huaxichenye@163.com (Y.C.)

[2] Department of Epidemiology and Health Statistics, West China School of Public Health and West China Fourth Hospital, Sichuan University, Chengdu 610041, China; sunyue9811@163.com (Y.S.); cairong.zhu@hotmail.com (C.-R.Z.)

[3] Department of Medicine and Life Science, Chengdu University of Traditional Chinese Medicine, Chengdu 611137, China; liuruizhi2003@163.com

[4] Sichuan Clinical Research Center of Biotherapy, Chengdu 610041, China

[5] Department of Oncology, The First People's Hospital of Ziyang, Ziyang 641300, China

[*] Correspondence: liujiyan1972@163.com; Tel.: +86-28-542-3261; Fax: +86-28-8542-3609

**Abstract:** Purpose: Immune-related pneumonitis (IRP) has attracted extensive attention, owing to its increased mortality rate. Conventional chemotherapy (C) has been considered as an immunosuppressive agent and may thus reduce IRP's risk when used in combination with PD-1/L1 inhibitors. This study aimed to assess the risk of IRP with PD-1/L1 inhibitors plus chemotherapy (I+C) versus PD-1/L1 inhibitors alone (I) in solid cancer treatment. Method: Multiple databases were searched for RCTs before January 2021. This NMA was performed among I+C, I, and C to investigate IRP's risk. Subgroup analysis was carried out on the basis of different PD-1/L1 inhibitors and cancer types. Results: Thirty-one RCTs (19,624 patients) were included. The I+C group exhibited a lower risk of IRP in any grade (RR, 0.60; 95% CI, 0.38–0.95) and in grade 3–5 (RR, 0.44; 95% CI, 0.21–0.92) as opposed to the I group. The risk of any grade IRP with PD-1 plus chemotherapy was lower than that with PD-1 monotherapy (RR, 0.50; 95% CI, 0.28–0.89), although grade 3–5 IRP was similar. There was no statistically meaningful difference in the risk of any grade IRP between PD-L1 plus chemotherapy and PD-L1 inhibitors monotherapy (RR, 0.95; 95% CI, 0.43–2.09) or grade 3–5 IRP (RR, 0.71;95% CI, 0.24–2.07). In addition, compared with the I group, the I+C group was correlated with a decreased risk in IRP regardless of cancer type, while a substantial difference was only observed in NSCLC patients for grade 3–5 IRP (RR, 0.39; 95% CI, 0.15–0.98). Conclusions: In comparison to PD-1/L1 inhibitor treatment alone, combining chemotherapy with PD-1/L1 inhibitors might reduce the risk of IRP in the general population. Furthermore, PD-1 inhibitors in combination with chemotherapy were correlated with a decreased risk of IRP compared to PD-1 inhibitor treatment alone. In contrast to the I group, the I+C group exhibited a lower risk of IRP, especially for NSCLC patients.

**Keywords:** immune-related pneumonitis; cancer; chemotherapy; programmed cell death-ligand 1 inhibitors; programmed cell death 1 inhibitors; network meta-analysis

## 1. Introduction

Cancer treatment has been radically altered by programmed cell death-ligand 1 (PD-L1) inhibitors and programmed cell death 1 (PD-1) over the last few decades [1]. The

FDA has approved many PD-1 and PD-L1 inhibitors for treating solid cancers. PD-1/L1 inhibitors alone (I) or combined with chemotherapy (I+C) appear to further improve clinical efficacy compared with conventional chemotherapy (C) [2–5].

PD-1/L1 inhibitors reactivate T cell-mediated anticancer immunity by inhibiting the PD-1 or PD-L1 immune checkpoint pathway [6]. In cases where cellular immunity is reactivated, these checkpoint inhibitors may cause inflammation-related side effects defined as immune-related adverse events (irAEs) [7]. Compared to the adverse events from chemotherapy, irAEs are uniquely organ-specific and can involve all organ systems, and a rash seems to be most frequent, followed by hypothyroidism and colitis [8]. Immune-related pneumonitis (IRP) is a relatively rare irAE, but still arouses great concern among clinicians due to its significant rate of treatment discontinuation and mortality. Previous studies have found that the incidence of IRP is between 3.8% and 9.6%, and that it also has a higher rate of IRP with use of PD-1/L1 combination therapy compared with PD-1/L1 monotherapy [9]. Traditional chemotherapy may facilitate an immunostimulatory impact through targeting cancer cells [10] and altering whole-body physiology [11]. In addition, numerous conventional chemotherapeutics can directly affect immune cells [12] and deplete immune effector cells and immunosuppressive cells. Chemotherapeutic drugs appear to have immunosuppressive impacts [13] due to their cytotoxic and cytostatic functions on different immune cell subpopulations. Thus, we can speculate that immunosuppression, caused by chemotherapy, may lower the risk of immune-mediated pneumonitis.

Earlier meta-analyses found that combining PD-1/L1 inhibitors with chemotherapy resulted in a reduced rate of grade 3–5 IRP compared to PD-1/L1 inhibitors alone in first-line therapy for advanced non-small-cell lung cancer (NSCLC) [12,13]. However, in terms of cancer types and different ICIs, no applicable trials comparing the incidence of IRP with a PD-1/L1 inhibitor, in combination with chemotherapy (I+C) to a PD-1/L1 inhibitor alone (I), have been conducted. Given the expanding population of cancer patients exposed to I+C treatment options, the number of IRP is expected to escalate in the future. Yet, whether this combination regimen increases the risk of IRP is still unclear. Employing a frequentist approach, we conducted a systematic and network meta-analysis (NMA) to investigate the risk of IRP among I+C combination and I monotherapy in a variety of cancer populations.

## 2. Methods and Materials

### 2.1. Search Methods and Study Selection

We conducted a comprehensive literature search to find existing clinical studies on PD-1 or PD-L1 inhibitors that recorded irAE. The search was carried out in PubMed, Embase, Web of Science, and Cochrane Library by applying the terms: cancer, tumor, neoplasm, pembrolizumab, nivolumab, durvalumab, atezolizumab, PD-1 inhibitors, and PD-L1 inhibitors. This systematic search was carried out on 1 October 2015, and ended with a final search for updates on 1 January 2021. Abstracts, as well as presentations from 2020 international conferences ESMO (the European Society for Medical Oncology) and ASCO (American Society of Clinical Oncology), were also included for additional eligible studies. The approval of the protocol in this study was granted by the West China Hospital of Sichuan University.

The Preferred Reporting Items for Systematic Reviews and Meta-analyses (PRISMA) guidelines and the Extension Statement for systematic reviews involving network meta-analyses were followed in this study [14]. Eligible RCTs were identified by reviewing titles, abstracts, full articles, and supplementary appendices. Studies eligible for inclusion met all of the following criteria: (1) cancer treatment randomized controlled trials (RCTs); (2) patients received PD-1/PD-L1 inhibitor therapy (on a minimum of one treatment arm); (3) RCTs gave the rate of IRP or the number of IRP and its grading; (4) all studies were published in English. We excluded RCTs involving PD-1/L1 combined with anti-cytotoxic T lymphocyte antigen-4 (CTLA-4) or other tyrosine kinase inhibitors (TKIs). In addition, RCTs, including PD-1/L1 monotherapy versus placebo or best supportive care and PD-1/L1 plus chemotherapy versus placebo or best supportive care, were also ineligible for this

study. All resources for each trial, including supplementary appendices, full-text articles, and conference abstracts (ASCO and ESMO), were used. The IRP grade was established using the National Cancer Institute's Common Terminology Criteria for Adverse Events. When multiple articles reporting on the same research population were found, the one with the most up-to-date and/or comprehensive irAE data was chosen. Publication selection, literature search, and extraction of data were conducted separately by Y.-X.L. and R.-Z.L. Disagreements were resolved by reaching an agreement through discussion with the help of a third experienced researcher (Y.S.).

### 2.2. Data Extraction and Quality Assessment

The trial name, phase, publication year, blinding method, PD-1 and PD-L1 inhibitors used, cancer type, median age with its corresponding range, numbers, and therapeutic interventions for experimental and control cohorts reported in the publication were extracted from each eligible study. The endpoints and median follow-up with the available corresponding duration for each included RCT were obtained. The rate or number of any grade IRP and grade 3–5 IRPs were retrieved. IRP was defined as immune-related pneumonitis or immune-mediated pneumonitis. Other pulmonary diseases classified as interstitial lung disease were not included in the study. The baseline data for each RCT were also obtained for analyses of study comparability.

### 2.3. Risk of Bias Assessment

By applying the research tool advised by the *Cochrane Collaboration Handbook* [14], the methodological quality of eligible studies was estimated based on the original research and the Supplementary Materials. This tool assessed the following aspects: selection bias (random sequence generation and allocation concealment), performance bias (blinding of participants and personnel), detection bias (blinding of outcome assessment), attrition bias (incomplete outcome data) performance bias (blinding of participants and personnel), detection bias (blinding of outcome assessment), attrition bias (incomplete outcome data), reporting bias (selective reporting), and other biases. Any disagreements were settled through agreement and discussion.

### 2.4. Outcome

By measuring the RR of the I+C group versus the I group among the overall population, this NMA primarily aimed to assess whether the addition of chemotherapy to PD-1/L1 monotherapy would decrease the risk of IRP. Subgroup analyses were performed for different ICIs and cancer types to investigate the risk of IRP between the I+C group and the I group.

### 2.5. Statistical Analysis

We conducted standard pairwise meta-analyses with the aid of Review Manager V.5.3 (https://training.cochrane.org/online-learning/core-software-cochrane-reviews/revman, accessed on 7 December 2021). The data were presented using a pooled risk ratio (RR) and a 95 percent confidence interval (CI). The heterogeneity of each pairwise comparison was assessed by Cochran's $I^2$ statistic, with values over 50% suggesting significant heterogeneity among research reports [15]. The random-effects model was the better option. Then, we conducted a network meta-analysis inside a frequentist framework with multivariate meta-analysis models, exploiting both the direct and indirect randomized evidence to determine the relative effects and rankings [16]. The frequentist approach was preferred based on the statistical expertise of our team, and results of the frequentist and Bayesian approaches were expected to be very similar. We reported on mixed evidence using RR for outcomes with 95% CI. If inconsistencies were not detected in the evidence, the consistency model was adopted to complete the relative effect of the included therapies. Nevertheless, the inconsistency model was employed as well. In addition, we also calculated the possibility of tolerability for each treatment and the surface under the cumulative

ranking curve (SUCRA) to compare the relative ranking probability for each treatment. Elevated SUCRA scores were associated with a greater IRP risk.

Inconsistency assessment was completed in two steps. First, the value of the inconsistency model test was calculated to verify the consistency assumption for the entire network. A *p*-value less than (<) 0.05 was considered an inconsistency in the entire network, and thus, the consistency model was not used; moreover, we examined inconsistency between the direct and indirect evidence by means of the loop-specific approach to calculate inconsistency factors (IF). When the IF value was close to "0" and its 95% CI contained "0", the direct and indirect comparison was considered as a strong consistency. Second, any inconsistencies were identified using node-splitting models. A *p*-value less than (<) 0.05 was defined as a conspicuous inconsistency.

To investigate the comparability of studies, baseline characteristics were compared using a Student's *t*-test. When *p* > 0.05, two groups were considered comparable at baseline.

## 3. Results

### 3.1. Eligible Studies and Characteristics

Figure 1 shows a PRISMA flow diagram depicting the study selection process. A total of 9205 records were retrieved through the initial database checks and reference searches, of which 274 potentially relevant studies were appropriate for full-text review. A total of 243 published articles were excluded because they were non-randomized controlled trials, the therapy regime involved TKIs or anti-CTLA4, or no usable data were reported. A total of 31 RCTs [2,17–33] about PD-1/L1 inhibitors, chemotherapy, or their combination for the treatment of solid cancer were included in this NMA.

The studies were divided into an I+C group, I group, and C group, while only three studies (KEYNOTE-062, KETNOTE-048, and IMvigor130) had both the I+C group and I group. Pembrolizumab (*n* = 9), pembrolizumab plus chemotherapy (*n* = 7), nivolumab (*n* = 6), atezolizumab (*n* = 2), atezolizumab plus chemotherapy (*n* = 7), durvalumab (*n* = 2), and durvalumab plus chemotherapy (*n* = 1) were among the PD-1 and PD-L1 inhibitors utilized. These trials involved the treatment of triple-negative breast cancer (TNBC) (*n* = 5), small cell lung cancer or non-small cell lung cancer (NSCLC) (*n* = 17), esophageal cancer, gastric or gastroesophageal cancer (ES, G/GEJ) (*n* = 3), urothelial cancer (UC) (*n* = 3), as well as other cancers (*n* = 3). Figure 2 shows network plots of the overall populations based on the interconnection of three treatments (I+C, I, and C groups). Moreover, we categorized the three treatment regimens into five subgroups according to different types of ICIs: chemotherapy, PD-1 monotherapy, PD-1 plus chemotherapy, PD-L1 monotherapy, and PD-L1 plus chemotherapy. Finally, we performed a comparison of the risk of IRP between the I+C group and the I group in different cancer populations. Network plots of subgroup analysis are presented in Figures S1 and S2.

The main demographic characteristics of all eligible studies are shown in Table 1 and Table S8. The findings of a comparability analysis at baseline are shown in Table S1. Table S1 shows no considerable differences between groups in terms of baseline data (*p* > 0.05).

### 3.2. Risk of Bias within Studies

Overall, eligible RCTs were regarded as having minimal risk for bias, except for detection bias, for which most of the trials had blinded assessments of outcomes, whereas the other trials were either not blinded or were open label. Due to a lack of grade 3–5 IRP data, four studies (IMpower133, KEYNOTE-062, IMvigor130, and KEYNOTE-181) were regarded as high risk in terms of attrition bias. The comprehensive appraisal findings are illustrated in Figures 3 and S3.

### 3.3. Risk of Bias within Studies

Overall, eligible RCTs were regarded as having minimal risk for bias, except for detection bias, for which most of the trials had blinded assessments of outcomes, whereas

the other trials were either not blinded or were open label. Due to a lack of grade 3–5 IRP data, four studies (IMpower133, KEYNOTE-062, IMvigor130, and KEYNOTE-181) were regarded as high risk in terms of attrition bias. The comprehensive appraisal findings are illustrated in Figures 3 and S3.

*3.4. Primary Outcome*

Results from three direct comparisons are shown in Figures S4 and S5. Compared with the I group, the I+C group may decrease the risk of IRP in any grade (RR, 0.80; 95% CI, 0.49–1.39). With the assistance of the consistency model, a forest plot for any grade of IRP in the three treatment groups illustrated that the I+C group exhibited a lower risk of IRP than the I group (RR, 0.60; 95% CI, 0.38–0.95). In line with any grade IRP, the I+C group (RR, 0.44; 95% CI, 0.21–0.92) also presented a reduced risk of grade 3–5 IRPs than the I group (Figure 4). The ranking probability based on the three groups is illustrated in Table S2. The I group was found to have the highest ranking (99.3), followed by the I+C group (50.7), and the C group (0.0) for the grade 1–5 IRPs in the overall population. Consistency was observed in the ranking between grade 3–5 IRPs and grade 1–5 IRPs from high to low: I group (99.1), I+C group (50.6), and C group (0.0).

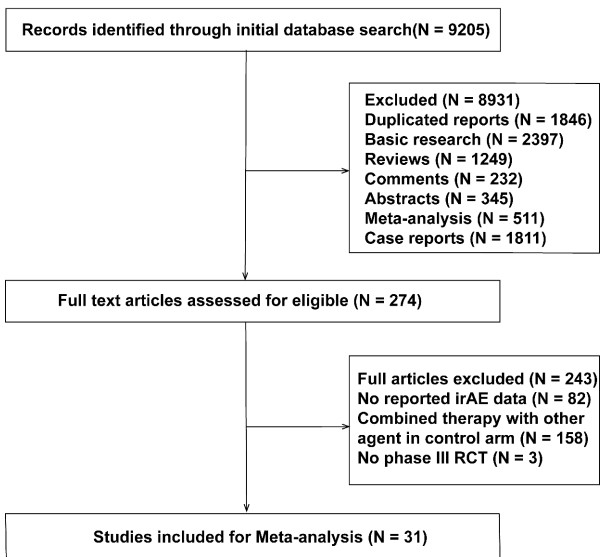

**Figure 1.** The study selection process for network meta-analysis on the occurrence of immune-related pneumonitis.

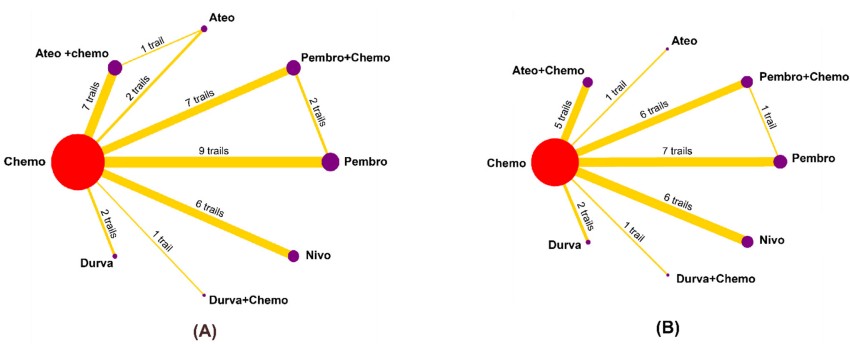

**Figure 2.** Networks established for comparisons based on all treatment groups: (**A**) for any grade IRP and (**B**) for grade 3 or higher IRPs. Each circular node represents a type of treatment. Red indicates chemotherapy. Other colors mean different PD-1/L1 inhibitors. The node size is proportional to the total number of patients receiving a treatment. Each line represents a type of head-to-head comparison. The width of lines is proportional to the number of trials. IRP: immune-related pneumonitis.

**Table 1.** Demographic characteristics of included random clinical trails (RCTs).

| Study Name | Year | Phase | Blind | History | Patients, No. | | Median Age | | Treatment | | Median Follow-Up Time | Endpoint | |
|---|---|---|---|---|---|---|---|---|---|---|---|---|---|
| | | | | | Experiment | Control | Experiment | Control | Experiment | Control | | Primary | Second |
| CASPIAN | 2021 | 3 | Double | SCLC | 268 | 269 | 62 (58–68) | 63 (57–68) | Durvalumab + (platinum + etoposide) | Platinum + etoposide | 25.1 (22.3–27.9) | OS | PFS, ORR |
| KEYNOTE-062 | 2020 | 3 | Partially | G/GEJ | 257 | 256 | 62 (22–83) | 61 (20–83) | Pembrolizumab + (cisplatin + fluo-rouracil/capecitabine q3w) | Pembrolizumab | 29.4 (22.0–41.3) | OS, PFS | ORR, DOR, safety, |
| IMpassion130 | 2020 | 3 | Double | TNBC | 451 | 451 | 55 (46–64) | 56 (47–65) | Atezolizumab + nab-paclitaxel | Placebo + nab-paclitaxel | 18.5 (9.6–22.8) | PFS, OS | ORR, DOR, QOF, |
| KEYNOTE-522 | 2020 | 3 | Double | TNBC | 784 | 390 | 49 (22–80) | 48 (24–79) | Pembrolizumab + (paclitaxel + carboplatin) | Paclitaxel + carboplatin | 15.0 (2.7–25.0) | pCR; EFS | pCR; safety |
| KEYNOTE-024 | 2019 | 3 | Open | NSCLC | 154 | 151 | 64 (33–90) | 66 (38–85) | Pembrolizumab (200 mg, q3w, up to 2 years) | Platinum-based (4–6 cycles) | 25.2 (20.4–33.7) | PFS | OS, ORR, DOR |
| KEYNOTE-604 | 2020 | 3 | Double | SCLC | 228 | 225 | 64 (24–81) | 65 (37–83) | Pembrolizumab + (platinum + etoposide) | Platinum + etoposide | 26.1 (16.1–30.6) | PFS, OS | ORR, DOR, safety |
| MYSTIC | 2020 | 3 | Open | NSCLC | 369 | 352 | 65 (28–84) | 64 (30–85) | Durvalumab (20 mg/kg, q4w) | Platinum-based | 30.2 (0.3–37.2) | OS, PFS | ORR, DOR, safety |
| KEYNOTE-407 | 2020 | 3 | Double | NSCLC | 278 | 281 | 65 (29–87) | 65 (36–88) | Pembrolizumab + (carboplatin + paclitaxel/nab-paclitaxel) | Placebo + (carboplatin + paclitaxel/nab-paclitaxel) | 14.3 (0.1–31.3) | OS, PFS | ORR, DOR, safety |
| KEYNOTE-045 | 2019 | 3 | Open | UC | 270 | 272 | 67 (29–88) | 65 (26–84) | Pembrolizumab (200 mg, q3w) | Paclitaxel, docetaxel, or vinflunine | 26 | OS, PFS | ORR, DOR, safety |
| IMpassion031 | 2020 | 3 | Double | TNBC | 165 | 168 | 51 (22–76) | 51 (26–78) | Atezolizumab plus (nab-paclitaxel + doxorubicin + cyclophosphamide) | Nab-paclitaxel + doxorubicin + cyclophosphamide | 20.6 (8.7–24.9) | pCR | EFS, OS, safety, tolerability |
| IMpower133 | 2020 | I/III | Double | SCLC | 201 | 202 | 64 (28–90) | 64 (26–87) | Atezolizumab plus (carboplatin + etoposide) | Carboplatin + etoposide | 23.1 (0–29.5) | OS, PFS | ORR, DOR, safety |

**Table 1.** *Cont.*

| Study Name | Year | Phase | Blind | History | Patients, No. | | Median Age | | Treatment | | Median Follow-Up Time | Endpoint | |
|---|---|---|---|---|---|---|---|---|---|---|---|---|---|
| | | | | | Experiment | Control | Experiment | Control | Experiment | Control | | Primary | Second |
| IMpower110 | 2020 | 3 | Open | NSCLC | 277 | 277 | 64 (30–81) | 65 (30–87) | Atezolizumab | Platinum-based | 13.4 (0–35) | OS | PFS, DOR, safety |
| KEYNOTE-010 | 2020 | II/III | Open | NSCLC | 690 | 343 | 63 (56–69) | 62 (56–69) | Pembrolizumab | Docetaxel | 42.6 (35.2–53.2) | OS, PFS | ORR, DOR, safety |
| KEYNOTE-189 | 2020 | 3 | Double | NSCLC | 410 | 206 | 65 (34–84) | 63 (34–84) | Pembrolizumab plus (pemetrexed + platinum) | Pemetrexed + platinum | 23.1 (18.6–30.9) | OS, PFS | ORR, DOR, safety |
| IMvigor130 | 2020 | 3 | Double | UC | 451 | 362 | 69 (62–75) | 67 (62–74) | Atezolizumab plus platinum-based chemotherapy | Atezolizumab | 11.8 (6.1–17.2) | OS, safety | ORR, DOR, PFS |
| IMpower130 | 2019 | 3 | Open | NSCLC | 483 | 240 | 64 (18–86) | 65 (38–85) | Atezolizumab plus (carboplatin + nab-paclitaxel) | Carboplatin + nab-paclitaxel | 18.5 (15.2–23.6) | PFS, OS | ORR, DOR |
| KEYNOTE-042 | 2019 | 3 | Open | NSCLC | 637 | 637 | 63 (57–69) | 63 (57–69) | Pembrolizumab | Platinum-based | 12.8 (6.0–20.0). | OS | ORR, DOR, PFS |
| KEYNOTE-061 | 2018 | 3 | Partially | G/GEJ | 296 | 296 | 62 (54–70) | 60 (53–68) | Pembrolizumab | Paclitaxel | 7.9 (3.4–14.6) | OS, PFS | ORR, DOR, safety |
| KEYNOTE-048 | 2019 | 3 | Open | HNSCC | 281 | 301 | 61 (55–68) | 62 (56–68) | Pembrolizumab plus (platinum + 5-fluorouracil) | Pembrolizumab | 11.5 (5.1–20.8) | OS, PFS | ORR, DOR, safety |
| IMpower132 | 2020 | 3 | Open | NSCLC | 292 | 286 | 64 (31–85) | 63 (33–83) | Atezolizumab plus (carboplatin/cisplatin + pemetrexed) | Carboplatin/cisplatin + pemetrexed | 14.8 (11.7–25.5) | OS, PFS | ORR, DOR, safety |
| IMpower131 | 2018 | 3 | Open | NSCLC | 343 | 340 | 65 (23–83) | 65 (38–86) | Atezolizumab + Carboplatin + Nab-Paclitaxel | Carboplatin + Nab-Paclitaxel | NR | PFS, OS | ORR, DOR, safety |
| KEYNOTE-355 | 2020 | 3 | Double | TNBC | 566 | 281 | 53 (44–63) | 53 (43–63) | Pembrolizumab + Chemotherapy (nab-paclitaxel/paclitaxel/ gemcitabine + carboplatin) | Nab-paclitaxel/paclitaxel/ gemcitabine + carboplatin | 25.9 (22.8–29.9) | Safety; PFS | ORR; DOR |

**Table 1.** *Cont.*

| Study Name | Year | Phase | Blind | History | Patients, No. | | Median Age | | Treatment | | Median Follow-Up Time | Endpoint | |
|---|---|---|---|---|---|---|---|---|---|---|---|---|---|
| | | | | | Experiment | Control | Experiment | Control | Experiment | Control | | Primary | Second |
| KEYNOTE-119 | 2021 | 3 | Open | TNBC | 312 | 310 | 50 (43–59) | 53 (44–61) | Pembrolizumab | Capecitabine, eribulin, gemcitabine, vinorelbine | 31.4 (27·8–34·4) | OS | PFS ORR DOR safety |
| DANUBE | 2020 | 3 | Open | UC | 346 | 344 | 67 (60–73) | 68 (60–73) | Durvalumab | Gemcitabine + cis-platin/carboplatin | 41.2 (37.9–43.2) | OS | ORR DOR |
| KEYNOTE-181 | 2021 | 3 | Open | ES | 314 | 314 | 63 (23-84) | 62 (24–84) | Pembrolizumab | Paclitaxel/docetaxel/irinotecan | 7.1 (0.5–31.3) | OS | PFS ORR |
| CheckMate 017 | 2015 | 3 | Open | NSCLC | 135 | 137 | 62 (39–85) | 64 (42–84) | Nivolumab | Docetaxel | NR | OS, ORR | PFS, Efficacy |
| CheckMate 057 | 2015 | 3 | Open | NSCLC | 292 | 290 | 61 (37–84) | 64 (21–85) | Nivolumab | Docetaxel | 12.2 (9.7–15.1) | OS | Efficacy PFS, ORR Efficacy |
| CheckMate 078 | 2015 | 3 | Open | NSCLC | 338 | 166 | 60 (27–78) | 60 (38–78) | Nivolumab | Docetaxel | 8.8 (0.2–21.1) | OS | PFS, ORR |
| CheckMate 026 | 2017 | 3 | Open | NSCLC | 271 | 270 | 63 (32–89) | 65 (29–87) | Nivolumab | Platinum doublet chemotherapy | 13.5 | PFS | OS |
| CheckMate 066 | 2015 | 3 | Double | MM | 210 | 208 | 64 (18–86) | 66 (26–87) | Nivolumab | Dacarbazine | 16.7 | OS | PFS |
| CheckMate 037 | 2015 | 3 | Open | MM | 272 | 133 | 59 (23–88) | 62 (29–85) | Nivolumab | Dacarbazine or paclitaxel + carboplatin | 5.3 (3.3–6.5) | ORR | PFS, OS |

Note: OS: overall survival; PFS: progression-free survival; ORR: objective response rate; DOR: duration of response; pCR: pathological complete response; EFS: event-free survival; SCLC: small cell lung cancer; G/GEJ: gastric and gastroesophageal junction cancer; TNBC: triple-negative breast cancer; NSCLC: non–small-cell lung cancer; UC: urothelial cancer; HNSCC: head and neck squamous cell carcinoma; ES: esophagus; NR: not reported.

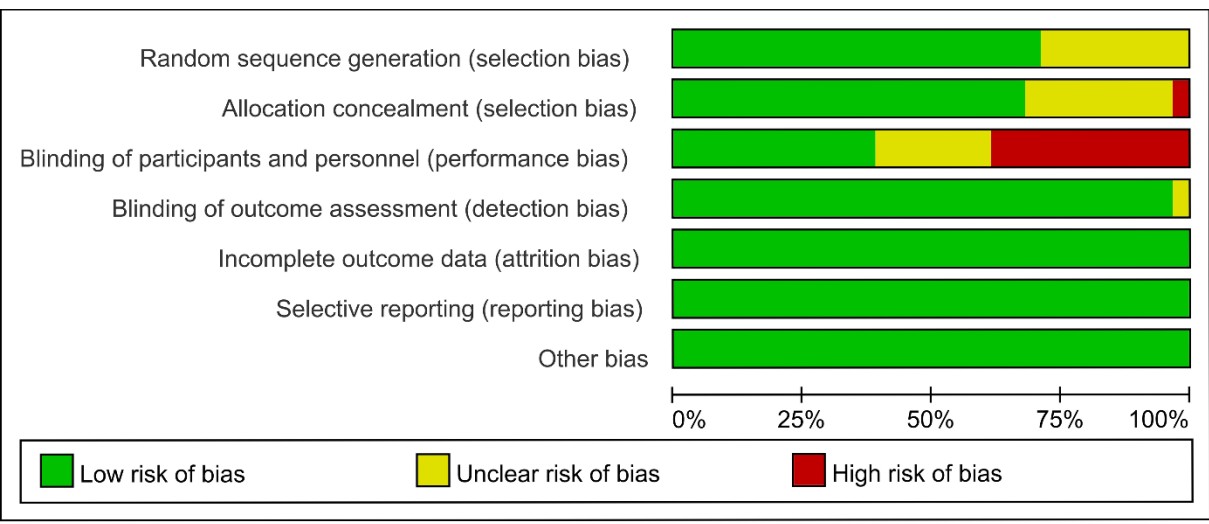

**Figure 3.** Risk of bias graph: review author's judgements presented as percentages.

*3.5. Subgroup Analysis of IRPs by the Different Types of ICIs*

We performed a comparison of the risk of IRP among different ICIs-combination regimes based on the consistency model. As shown in Figure 5, in contrast with PD-1 monotherapy, a reduced risk of any grade of IRP was detected in the PD-1 plus chemotherapy (RR, 0.50; 95% CI, 0.28–0.89). However, a comparable result was found in the PD-L1 plus chemotherapy versus PD-L1 monotherapy in any grade of IRP (RR, 0.95; 95% CI, 0.43–2.09). The corresponding ranking of these five treatment groups from highest to lowest was: PD-1 monotherapy (97.8), PD-L1 monotherapy (52.4), PD-1 plus chemotherapy (52.4), PD-L1 plus chemotherapy (47.3), and chemotherapy (0.0) (Table S3). According to the consistency model, there were fewer differences between the five treatment groups for the risk of grade 3–5 IRPs (Figure 6). No substantial differences were found between PD-1 monotherapy and PD-1 plus chemotherapy. The probability rankings from highest to lowest were PD-1 monotherapy (92.5), PD-L1 monotherapy (65.7), PD-1 plus chemotherapy (52.9), PD-L1 plus chemotherapy (34.0), and chemotherapy (4.8) (Table S3).

*3.6. Subgroup Analysis of IRP by Caner Type*

Based on the type of cancer treated in the RCTs, we classified the 31 studies into 4 different categories. As shown in Table S4, the I+C group had a lower risk in comparison with the I group for any grade of IRP, regardless of cancer type, even though the results did not seem statistically significant: NSCLC (RR, 0.44; 95% CI, 0.19–1.01), TNBC (RR, 0.25; 95% CI, 0.00–14.09), UC (RR, 0.23; 95% CI, 0.04–1.33) and ES, G/GEJ (RR, 0.91; 95% CI, 0.36–2.27). The SUCRA value for cancer types among these groups is presented in Table S4. Higher SUCRA scores were associated with higher risk of IRP. Similarly, no statistically significant difference was found in the risk of grade 3–5 IRPs between I+C group and I group for TNBC (RR: 0.46; 95% CI, 0.02–12.13). However, the I+C group had a lower risk of grade 3–5 IRPs than the I group (RR, 0.39; 95% CI, 0.15–0.98) for NSCLC. In the SUCRA analysis, the I+C group (50.7) was better tolerated in comparison with the I group (99.3) with regard to grade 3–5 IRPs for NSCLC.

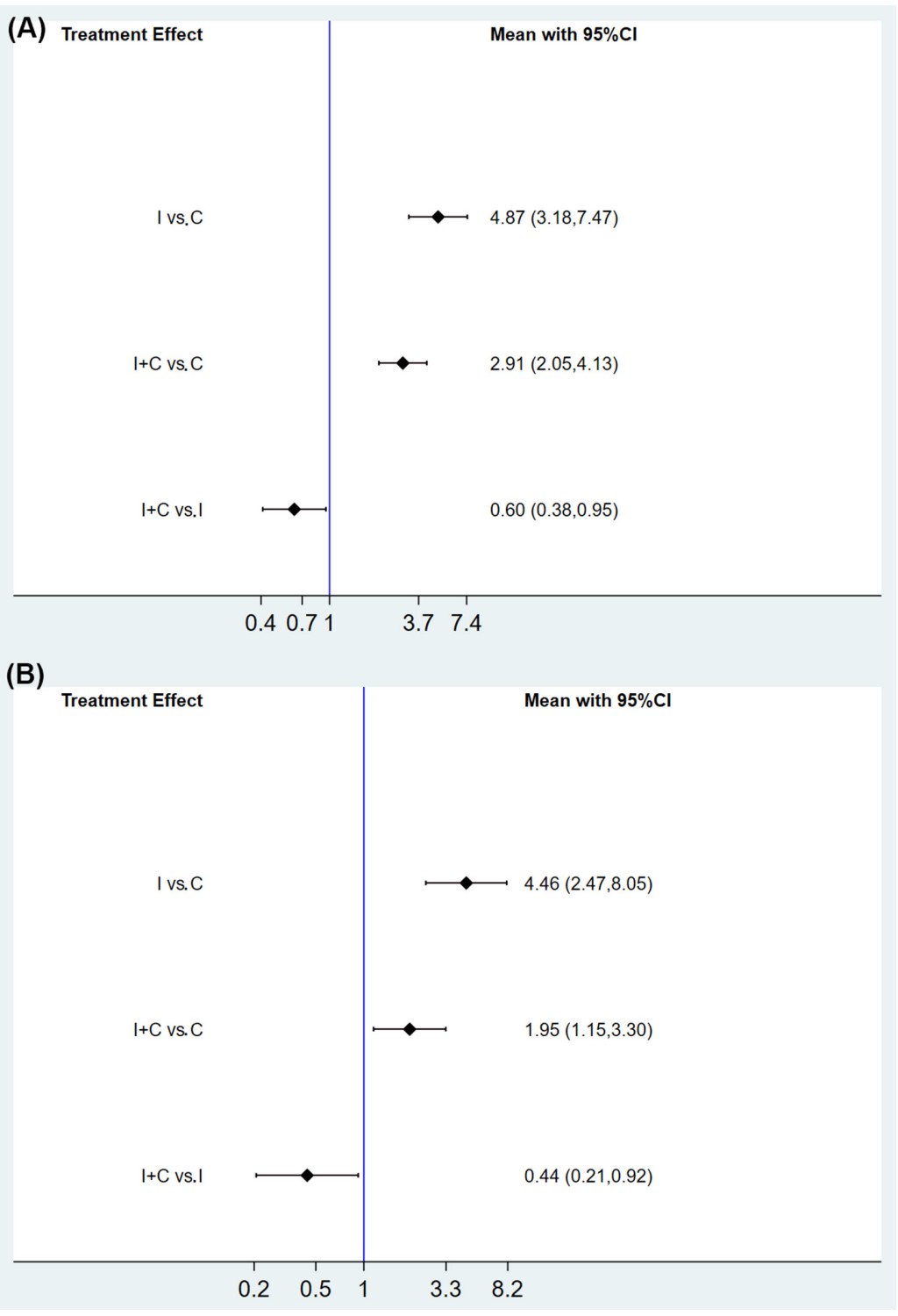

**Figure 4.** Comparative different treatments for the risk of any grade IRP (**A**) and grade 3–5 IRPs (**B**) in the overall population, based on the consistency model. IRP, immune-related pneumonitis; I+C group, PD-1/L1 inhibitors plus chemotherapy; I group, PD-1/L1 inhibitors monotherapy; C group, chemotherapy.

| PD–1 monotherapy | 0.50 (0.28–0.89) | 0.49 (0.20–1.19) | 0.46 (0.23–0.95) | 0.16 (0.10–0.28) |
|---|---|---|---|---|
| 2.02 (1.12–3.62) | PD–1+Chemotherapy | 0.99 (0.41–2.40) | 0.93 (0.46–1.91) | 0.33 (0.20–0.55) |
| 2.04 (0.84–4.94) | 1.01 (0.42–2.46) | PD–L1 monotherapy | 0.95 (0.43–2.09) | 0.33 (0.16–0.69) |
| 2.16 (1.05–4.42) | 1.07 (0.52–2.19) | 1.06 (0.48–2.34) | PD–L1+Chemotherapy | 0.35 (0.21–0.59) |
| 6.11 (3.64–10.26) | 3.03 (1.82–5.05) | 2.99 (1.44–6.22) | 2.83 (1.71–4.68) | Chemotherapy |

**Figure 5.** Different ICI comparisons for all grades of IRP based on network consistency model. Comparisons should be read from left to right. RRs for comparisons are in the cell in common between the column-defining and row-defining treatment, and bold numbers with darker backgrounds are statistically significant; values that were borderline significant are in a lighter shade. An RR > 1 means the treatment in the top left is worse. The numbers in parentheses represent 95% confidence intervals.

| PD–1 monotherapy | 0.45 (0.19–1.08) | 0.60 (0.14–2.58) | 0.32 (0.11–0.92) | 0.20 (0.10–0.39) |
|---|---|---|---|---|
| 2.22 (0.93–5.32) | PD–1+Chemotherapy | 1.33 (0.31–5.79) | 0.71 (0.24–2.07) | 0.44 (0.22–0.87) |
| 1.67 (0.39–7.18) | 0.75 (0.17–3.25) | PD–L1 monotherapy | 0.53 (0.11–2.47) | 0.33 (0.09–1.21) |
| 3.14 (1.09–9.10) | 1.41 (0.48–4.14) | 1.89 (0.40–8.81) | PD–L1+Chemotherapy | 0.62 (0.27–1.43) |
| 5.04 (2.59–9.82) | 2.27 (1.15–4.49) | 3.03 (0.83–11.10) | 1.60 (0.70–3.67) | Chemotherapy |

**Figure 6.** Different ICI comparisons for grade 3–5 IRPs based on network consistency model. Comparisons should be read from left to right. RRs for comparisons are in the cell in common between the column-defining and row-defining treatment, bold numbers with darker background are statistically significant; values that were borderline significant are in a lighter shade. An RR > 1 means the treatment in the top left is worse. The numbers in parentheses represents 95% confidence intervals.

*3.7. Heterogeneity, Inconsistency, and Publication Bias*

Three feasible pairwise comparisons with heterogeneity evaluations are presented in Figures S4 and S5. A low heterogeneity in all grade IRP and grade 3–5 IRPs was demonstrated by all the direct comparisons. Moreover, these three comparisons results also showed significant consistency in terms of tendency concerning the comparable NMA outcomes. Tables S5–S7 illustrate the findings of the inconsistency estimates. There was no significant inconsistency in the fit of the consistency models nor in the node splitting analyses. This NMA did not reveal any obvious publication bias. In general, funnel plots were close to the zero line and were roughly symmetrical (Figure S6).

## 4. Discussion

Immune check inhibitors (ICIs) have recently been recognized as one of the most important therapeutic options for solid cancer. Nonetheless, their wide application has induced a perceptible growth in IRPs. Previously, binary meta-analyses have confirmed that IRP incidence is elevated with ICIs-based combination therapy as compared to monotherapy and could differ across diverse types of cancer and ICIs [9]. Furthermore, some NMAs associated only with advanced lung cancer have assessed the IRP risk in different ICI-based

therapies. These studies were restricted to the small sample size, necessitating the pooling of all kinds of treatment. With more I+C therapeutic approaches gaining approval for advanced solid cancer, a thorough NMA was required to systematically evaluate and perform comparisons of the risk of IRP for the I+C regimens. This NMA consisted of 31 head-to-head phase III RCTs (19,624 patients) and was the first and largest NMA comparing the risk of IRP across ICI-based treatments for solid cancers.

Our findings illustrated a significantly lower risk of IRP with the I+C group than the I group for any grade and grade 3–5, in line with a recent NMA reported by Chen, et al. [34] and an indirect meta-analysis reported by Wang, et al. [35]. Their results showed that I+C regimes were related to a lower risk of any grade and grade 3–5 IPR than I monotherapy for patients with advanced lung cancer. One potential explanation for the reduced IRP risk could lie in the fact that chemotherapy was comprised of cytotoxic agents and considered as an immunosuppressive agent when conventional chemotherapeutics were used in combination with ICIs [13,36]. Several prospective studies have demonstrated that conventional chemotherapy significantly depleted all lymphocyte subpopulations [37], whereas only CD4+ T cell numbers in blood remained depleted six months after completion of chemotherapy [38]. Additionally, Suresh, K. and colleagues [39] found that bronchoalveolar lavage (BAL) samples from IRP patients exhibited increasing lymphocytosis that was predominantly composed of CD4+ T cells. Instead, Karpathiou, et al. observed that CD4+ T cells in the BAF fluid of healthy lung in patients treated with standard chemotherapy regimens decreased after chemotherapy [40]. Therefore, chemotherapy may affect and alter the immune environment of lung parenchyma and reduce the risk of IRP when in combination with ICIs. Additionally, since binding pretreatment was used for an antiallergy and antiemetic purposes and commonly in chemotherapy regimens that contained pemetrexed, taxanes, and platinum, another crucial factor attributed to the reduced risk of IRP was corticosteroid administration. It is well-recognized that corticosteroids have the potential to induce significant immunosuppressive and anti-inflammatory effects [41], as well as play a leading role in IRP management, according to ESMO, ASCO, and NCCN guidelines [42–44]. Unfortunately, the precise mechanism of immunosuppression from cytotoxic agents and corticosteroids remains unclear and requires further exploration.

PD-1 inhibitors plus chemotherapy were found to result in a statistically significant lower risk of any grade IRP compared to PD-1 inhibitor monotherapy in our NMA. PD-L1 plus chemotherapy had a risk that was statistically similar to that of PD-L1 monotherapy for any grade of IRP. One possible reason for these results was that PD-L1 inhibitors had a lower incidence of any grade IRP than PD-1 inhibitors (1.3% vs. 3.6%) [45]. This trend could also be observed in the ranking in any grade IRP or grade 3–5 IRPs. Therefore, it was difficult for the addition of chemotherapy to PD-L1 inhibitors monotherapy to further decrease the risk of IRP. In addition, PD-1 inhibitors may stimulate the integration of PD-L2 and repulsive guidance molecule b (RGMb), which increases the number of resident T cells in the lung and eventually results in IRP [46,47]. Previous studies have confirmed that conventional chemotherapy may deplete all lymphocyte subpopulations, including T cells. Consequently, combining chemotherapy with PD-1 inhibitor monotherapy might decrease the risk of IRP. However, PD-L1 inhibitors had no ability to disturb the balance in PD-L2 interactions with RGMb. The risk of IRP does not decrease when chemotherapy is used in combination with PD-L1 inhibitors.

The incidence of IRP was varied in different cancer types [9,48] and was demonstrated in previous studies as well. In this case, IRP seemed to have a higher likelihood of occurrence in NSCLC patients for those who usually had previously received chest radiotherapy or had a history of chronic obstructive pulmonary disease [49]. In addition, some authors have considered that various pathological types may have an effect on the occurrence of IRP, although in the same kind of cancer. A retrospective study illustrated that in NSCLC patients, a histology of adenocarcinoma tumors was related to reduced risk of IRP as opposed to those with non-adenocarcinoma tumor histology [50]. However, results from our NMA analysis confirmed that the I+C group exhibited a reduced risk of IRP as opposed

to the I group, either in NSCLC or in other kinds of cancers. The results, however, except for NSCLC patients, had no significant statistical difference between the I+C group and I group. This trend also could be seen in terms of the ranking, both in any grade IRP and grade 3–5 IRPs. Considering that there were a limited number of studies providing direct comparison between the I+C group and the I group, whether in NSCLC or other cancer types, these findings should be interpreted cautiously. Further high-quality RCTs are required to explore the incidence of IRP between the I+C group and the I group.

## 5. Limitations

This network meta-analysis has some limitations. First, this research only contained patients who were explicitly listed as having pneumonitis and did not include those noted as having interstitial lung diseases and pneumonia in the data extraction process. The current paradigms for the diagnosis of IRP are largely based on some clinical symptoms and radiographic infiltrates, primarily due to the lack of a consensus on IRP diagnostic criteria. Some other types of pneumonitis, such as COVID-19 and radiation pneumonitis, have similar clinical characteristics and imaging features to IRP. Distinguishing between IRP and other pneumonitis types is a diagnostic challenge. In addition, IRP reporting was voluntary, and the detection of IRP may not be precise. We suspect that the occurrence of IPR was underestimated. Suresh et al. recently found that IRP incidence was 19% in NSCLC patients receiving ICI treatments in real-world settings [36], which was much higher than the incidence of around 3% to 5% based on RCTs [33]. Therefore, our findings need to be validated in the real world. Additionally, there may exist a trend that the incidence of IRP reporting increases over time. Second, high-risk factors, which may include a history of previous smoking status and chest radiotherapy, were not collected during the process of performing our NMA, as none of the included RCTs, except lung cancer, has reported high-risk factors. Lastly, there were only three RCTs where I+C and I regimens were directly compared. Although this study has several limitations, we found a number of useful results that may help clinicians decide on appropriate ICI-based therapies. In the near future, we expect to see an increased number of double-blind RCTs, emphasizing head-to-head comparisons that focus on PD-1/L1 in combination with chemotherapy versus PD-1/L1 monotherapy. Notably, findings from real-world investigations are urgently required to corroborate our results.

## 6. Conclusions

In conclusion, PD-1/L1 inhibitors in combination with chemotherapy exhibited a lower risk of IRP both in any grade and grades 3–5 than PD-1/L1 monotherapy. In terms of ICI type, PD-1 inhibitors in combination with chemotherapy were associated with lower risk than PD-1 inhibitors monotherapy. No obvious statistical difference was discovered between PD-L1 inhibitors in combination with chemotherapy and PD-L1 inhibitor monotherapy. Moreover, compared with the I group, a decreased risk of IRP was found in the I+C group in spite of cancer type, particularly for NSCLC patients.

**Supplementary Materials:** The following are available online at https://www.mdpi.com/article/10.3390/curroncol29010025/s1, Table S1: Demographic Characteristics of the Patients at Baseline; Table S2: Value of SUCRA and mean rank in overall population. Higher SUCRA scores correlated with a higher risk of IRP; Table S3: Value of SUCRA and mean rank for different ICIs comparison. Higher SUCRA scores correlated with a higher risk of IRP; Table S4: The value of SCURA for 3 group and mixed RR that I+C group Versus I group based on cancer type; Table S5: The value of P for inconsistency model test and IF for loop inconsistency in overall population and subgroup analysis; Table S6: Results of node-splitting analysis for the assessment inconsistencies for the IRP in cancer type; Table S7: Results of node-splitting analysis for the assessment inconsistencies for the IRP in different ICIs group; Table S8: Supplementary basic characteristic for eligible RCTs. Figure S1: Network plots of subgroup analysis based on different ICIs (A) any grade IRP, (B) grade 3–5 IRP; Figure S2: Network established for comparisons based on cancer type. (A) any grade IRP and (B) grade 3–5 IRP in NSCLC, (C) any grade IRP and grade 3–5 IRP in TNBC, (D) any grade IRP in UC, (E)

any grade IRP in G/GEJ; Figure S3: Risk of bias summary: review author's judgements about each risk of bias item for each included study; Figure S4: Forest plots and pairwise meta-analysis of direct comparisons for the risk of any grade IRP. ICIs, immune checkpoint inhibitors; IRP, immune-related pneumonitis; Figure S5: Forest plots and pairwise meta-analysis of direct comparisons for the risk of grade 3 or higher IRP. ICIs, immune checkpoint inhibitors; IRP, immune-related pneumonitis; Figure S6: Publication bias for (A) any grade IRP and (B) grade 3–5 IRP in all population.

**Author Contributions:** Conceptualization, J.-Y.L. and Y.-X.L.; methodology, Y.-X.L., Y.S. and C.-R.Z.; software, M.-Y.Z.; validation, R.-Z.L., J.Z. and Y.C.; formal analysis, Y.-X.L.; investigation, Y.S.; resources, Y.-Q.W.; data curation, Y.-X.L. and R.-Z.L.; writing—original draft preparation, Y.-X.L.; writing—review and editing, J.-Y.L.; visualization, Y.-W.Z.; supervision, K.C.; project administration, Y.-X.L. All authors have read and agreed to the published version of the manuscript.

**Funding:** This study was supported by the Sichuan Science and Technology Department Key Research and Development Project (2019YFS0539), the 1.3.5 Project for Disciplines of Excellence, West China Hospital, Sichuan University (ZYJC18022), and National Natural Science Foundation of China (No.81572380).

**Institutional Review Board Statement:** Not applicable.

**Informed Consent Statement:** Not applicable.

**Data Availability Statement:** Data are contained within the article and Supplementary Materials.

**Conflicts of Interest:** The authors declare no conflict of interest.

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
