# Peer review of "Immune-Related Pneumonitis Was Decreased by Addition of Chemotherapy with PD-1/L1 Inhibitors: Systematic Review and Network Meta-Analysis of Randomized Controlled Trials (RCTs)"

_curroncol, doi:10.3390/curroncol29010025_

Round 1

Reviewer 1 Report

The study by Long et al is interesting an useful for the clinical practice of immunotherapy. It also provides evidence that the mild immunosuppression conferred by chemotherapy reduces immunotherapy related toxicities.

Aside to statistics, and since the paper is addressed to clinicians, the absolute numbers of patients with  ir-pneumonitis should be reported. For example Table 1 should be accompanied by a relevant additional table where the number of patients with pneumonitis and the total population of each treatment group, of the 31  studies included in the analysis, will be reported.

Author Response

We deeply appreciate the reviewer’s suggestion. We have tried our best to revise the manuscript according to your kind and construction comments and suggestions. We reported the number of patients with pneumonitis and the total population of each treatment group of the included 31 RCTs in table S8. Thank you very much for your comments and suggestions.

Reviewer 2 Report

In this paper, the authors are trying to analyze the risk of IRP with PD-1/L1 inhibitors alone and in combination in cancer treatment. They performed clinical trials to investigate IRP’s threat by dividing the groups into subgroups and found out PD-1/L1 inhibitors in combination with chemotherapy exhibited a lower risk of IRP both in any grade and grade 3-5 than PD-1/L1monotherpay.
The manuscript is submitted under the "Systematic Review" section but the manuscript is confusing between the original research article and review article: Line-18: "This study aimed to assess the risk of IRP with PD-1/L1 inhibitors plus chemotherapy (I+C) versus PD-1/L1 19 inhibitors alone (I) in solid cancer treatment".
In the introduction section, the authors do a great job of introducing everything but Immune-related pneumonitis.
The data information is incomplete. The name of the studies are provided in reference to the database collected but that is not enough to linkage the regenerate the results. The resource of the data with web-link or study id number should be provided together with the final data file generated by pooling the data sets for the analysis i.e. data mentioned in Table 1. .
The draft needs careful reading in order to eliminate typographical mistakes and use of acronyms.
Challenges and limitations with existing research in this area should be highlighted in the introduction section.
The scientific results are manipulated for example: "Figure 4. Comparative different treatments for the risk of any grade IRP(A) and grade 3-5 IRP (B),.." the x-axis is entity random.
Figures need significant improvement in its quality
Addition of a graphical abstract can add value to the manuscript.
The discussion section is not well structured and needs to be modified.  
Table 2. can be converted to a heatmap figure. 

Reviewer 3 Report

Very interesting and well performed systematic review and meta-analysis about decreased rate of immune-related pneumonitis by adding chemotherapy to ICI therapy.

A very few suggestions below:

  • Introduction: pneumonitis is not the only adverse effect of ICI therapy. Other effects are known such as colitis, hypophysitis and so on. Please provide an overview.
  • Introduction\limitations: the diagnosis of ICI pneumonits is challenging and can overlap with other conditions including COVID-19 and radiation pneumonitis (for references see "doi: 10.3390/cancers13040652"). Please provide an overview
  • Methods\results: did you evaluate the rate of patients under chronic corticosteroids therapy across the included studies? Any effect on the rate of pneumonitis? (ICI + CT + steroids = less rate?). This would be a nice addition to your analysis.

Round 2

Reviewer 3 Report

Nice job.